# Enantioselective cyanation via radical-mediated C–C single bond cleavage for synthesis of chiral dinitriles

Tao Wang[1], Yi-Ning Wang[1], Rui Wang[1], Bo-Chao Zhang[1], Chi Yang[1], Yan-Lin Li[1] & Xi-Sheng Wang [1]*

Ring-opening reaction via selective cleavage of C–C bond is known as a powerful strategy for construction of complex molecules. Complementary to the ionic process focusing on mostly small ring systems, radical-mediated C–C bond cleavage offers a solution for further diverse enantioselective functionalization benefited from its mild conditions, whereas such asymmetric transformations are still limited to three-membered rings so far. Herein, we describe radical-mediated ring-opening and enantioselective cyanation of four- and five-membered cycloketone oxime esters to access chiral 1,5- and 1,6-dinitriles. Employment of dual photoredox/copper catalysis is essential for the asymmetric ring-opening cyanation of cyclopentanone oxime esters. Both reactions proceed under mild conditions giving chiral dinitriles in high yields and enantioselectivity with low catalyst loading and broad substrate scope. The products dinitriles can be converted to valuable optically active diamides and diamines. Mechanistic studies indicate that the benzylic radical generated via C–C single bond cleavage is involved in the catalytic cycle.

---

[1] Hefei National Laboratory for Physical Sciences at the Microscale and Department of Chemistry, Center for Excellence in Molecular Synthesis of CAS, University of Science and Technology of China, 96 Jinzhai Road, 230026 Hefei, Anhui, P. R. China. *email: xswang77@ustc.edu.cn

Synthesis of complex natural products or functional molecules in an efficient and quick manner always represents a major challenge in organic chemistry. C–C single bond is basic element and linkage in organic molecules, selective cleavage and functionalization of which is of great significance and provides a straightforward and atom-economical method to access target molecule[1–6]. Indeed, ring-opening reaction via selective cleavage of C–C bond has long been used as a powerful strategy in retrosynthetic analysis to build up complexity of a molecule in a single step[7–10]. Due to the thermodynamic stability of C–C single bond, high temperature is usually required to enhance the reactivity, which is normally considered unfavorable to achieve high enantioselectivity. Thus, asymmetric reactions induced by ring-opening C–C bond cleavage still remain challenging since there is requirement of a balanced relationship between selectivity and reactivity in such transformations[11]. To date, the known asymmetric reactions via ring-opening C–C bond cleavage are mainly studied in the field of transition-metal catalysis[12–19] (Fig. 1a), Lewis acid catalysis[20–23], and organocatalysis[24–27] (Fig. 1b). However, most of these reactions via ionic pathway focused on C–C bond cleavage of small ring systems (3 or 4-membered rings), in which intrinsic strain release is exploited as a major driving force. Complementary to the ionic processes, radical-mediated C–C bond cleavage offers an efficient pathway for further asymmetric functionalization due to the mild conditions and reaction diversity[28–31]. However, such asymmetric reactions involving radical-promoted ring-opening C–C bond cleavage are still limited to three-membered rings so far (Fig. 1c)[32–38].

Dinitriles, known as synthetically useful motifs in many pharmaceuticals or biologically active compounds[39–44], are also widely used in polymer materials, enzyme catalysis and organic synthesis[45–50]. For instance, the most widely used dinitrile, adiponitrile, serves as a key precursor to produce the polymer nylon-66, which is used as fibers for clothing, tire cords, carpets, conveyor belts, and brushes due to its excellent dyeability and durability[51]. Despite the widespread application, the efficient synthesis of chiral dinitriles is still underdeveloped and remains as a big challenge. To our best knowledge, the only catalytic method to make chiral dinitriles is the asymmetric addition of cyano-containing building blocks to $\alpha$,$\beta$-unsaturated nitriles[52–55]. According to the pioneering discoveries of Zard[56,57] and recent works from other groups[58,59], iminyl radical-mediated ring-opening of cyclic oxime derivatives is an efficient pathway to cleave C–C single bond via $\beta$-scission. As such transformation generates cyanoalkyl radical in situ, we envision this radical could be further trapped by chiral transition-metal catalyst, followed by reductive elimination from the metal cyanide complex to furnish corresponding optically active dinitrile[60–66].

Herein, we report enantioselective ring-opening cyanation for facile synthesis of chiral 1,5- and 1,6-dinitriles (Fig. 1d). It is worthy of note that C–C single bond cleavage and asymmetric radical cyanation of less strained 5-membered cycloketone oxime ester can be expediently achieved by merging photoredox catalysis with copper catalysis. Under mild conditions, both asymmetric transformations demonstrate high catalytic reactivity, excellent enantioselectivity, low catalyst loading and broad substrate scope. The key to success is the capture of the cyanoalkyl radical in situ generated via ring-opening C–C cleavage with chiral copper catalyst resulting in an enantioselective cyanation through stereoselective reductive elimination.

## Results

**Optimization of the asymmetric ring-opening cyanation.** Our studies commenced with cyclopentanone oxime ester **1a** used as the pilot substrate, and trimethylsilyl cyanide (TMSCN) as the coupling partner in the presence of a catalytic amount of Cu(MeCN)$_4$PF$_6$ (10 mol%) at room temperature. To our delight, the desired ring-opening cyanation product **2a** was afforded with 91% *ee* when chiral bis(oxazoline) **L1** was used as the ligand, albeit in a rather low yield (23%, entry 1, Table 1). Considering high energy requirement of ring-opening due to the thermodynamic stability of C–C single bond in less strained

**Fig. 1** Enantioselective functionalization via ring-opening C–C bond cleavage. **a** Transition-metal-catalysis: normally strained rings. **b** Lewis acid-mediation or organocatalysis: strained rings only. **c** Radical-mediation: three-membered rings only. **d** This work: asymmetric radical transformations of four- and five-membered rings.

**Table 1 Optimization of reaction conditions[a].**

1a, R = p-CF$_3$C$_6$H$_4$

n = 1, **L1**
n = 2, **L2**
n = 3, **L3**

R = Me, **L4**
R = Bn, **L5**

**L6**

| entry | [Cu]/mol% | ligand | Solvent | T/oC | Yield (ee)/% |
|---|---|---|---|---|---|
| 1[b,c] | 10 | **L1** | DCM | RT | 23 (91) |
| 2[b,c] | 10 | **L1** | DCM | 50 | 63 (88) |
| 3[c,d] | 5 | **L1** | DCM | 50 | 55 (90) |
| 4[c,e] | 15 | **L1** | DCM | 50 | 58 (88) |
| 5 | 2 | **L1** | DCM | RT | 71 (91) |
| 6 | 2 | **L1** | CH$_3$CN | RT | 70 (91) |
| 7 | 2 | **L1** | Acetone | RT | 24 (93) |
| 8 | 2 | **L1** | Benzene | RT | 81 (87) |
| 9 | 2 | **L1** | PhCl | RT | 80 (87) |
| 10 | 2 | **L1** | DMF | RT | 73 (91) |
| 11 | 2 | **L1** | DMAc | RT | 73 (92) |
| 12 | 2 | **L2** | DMAc | RT | 68 (88) |
| 13 | 2 | **L3** | DMAc | RT | 70 (86) |
| 14 | 2 | **L4** | DMAc | RT | 58 (75) |
| 15 | 2 | **L5** | DMAc | RT | 40 (73) |
| 16 | 2 | **L6** | DMAc | RT | 56 (-43) |
| 17[f] | 2 | **L1** | DMAc | RT | 78 (92) |
| 18[g] | 2 | **L1** | DMAc | RT | 12 (90) |
| 19[h] | 2 | **L1** | DMAc | RT | 9 (91) |
| 20[i] | 0 | **L1** | DMAc | RT | 14 (0) |

Ir(ppy)$_3$ Tris(2-phenylpyridine)iridium, DCM Dichloromethane, DMF N,N-Dimethylformamide, DMAc N,N-Dimethylacetamide, RT room temperature
[a]Reaction conditions: **1a** (0.10 mmol), TMSCN (3 equiv), Cu(MeCN)$_4$PF$_6$ (2 mol%), ligand (3 mol%), Ir(ppy)$_3$ (0.5 mol%) in solvent (1.0 mL) at RT for 36 h under the irradiation of 5 W blue LEDs
[b]Cu(MeCN)$_4$PF$_6$ (10 mol%), **L1** (12 mol%)
[c]24 h
[d]Cu(MeCN)$_4$PF$_6$ (5 mol%), **L1** (6 mol%)
[e]Cu(MeCN)$_4$PF$_6$ (15 mol%), **L1** (18 mol%)
[f]TMSCN (1.5 equiv)
[g]In dark
[h]No Ir(ppy)$_3$
[i]No Cu(MeCN)$_4$PF$_6$

five-membered ring, we elevated the temperature to 50 °C to facilitate the process, and the yield was improved markedly to 63%, however expectedly, with a relatively lower *ee* (88%, entry 2). While decreasing catalyst loading resulted in significant recovery of **1a**, the increase of copper loading could not further improve the yield because of the side reactions involved at this temperature (entries 3–4). To further enhance the reactivity, photoredox catalysis was merged with copper catalytic system to provide extra energy. We reckoned that such system could react at lower temperature to inhibit side reactions and simultaneously improve the enantioselectivity. As expected, addition of Ir(ppy)$_3$ as the photocatalyst under the irradiation of 5 W blue LEDs at room temperature could raise the yield to 71% with 91% *ee*, and the copper catalyst loading could be reduced to only 2 mol% (entry 5). Moreover, a careful solvent screening (entries 5–11) indicated DMAc afforded a slighter higher yield (73%) and *ee* (92%, entry 11). Additionally, further screening of bis(oxazoline) ligands revealed that **L1** was the optimal ligand for this asymmetric transformation (entries 11–16). To our satisfaction, the

yield could be further improved to 78% when the amount of TMSCN was reduced to 1.5 equiv (entry 17). Finally, the control experiments showed only trace amount of the desired dinitrile **2a** was obtained in the absence of light, Ir(ppy)$_3$ or Cu(MeCN)$_4$PF$_6$ (entries 18–20), which clearly indicated that photoredox catalytic system and copper catalysis were both essential for this highly efficient transformation.

**Substrate scope of the cyclopentanone oxime esters 1**. With the optimized conditions in hand, we next started to expand the substrate scope of this asymmetric ring-opening cyanation. As shown in Fig. 2, a variety of cyclopentanone oxime esters **1** installed with various substituted aromatic rings were compatible with this protocol, affording chiral 2-arylhexanedinitriles **2** in good to excellent yields and enantioselectivity. The substituent effect of the aryl rings on oxime esters **1** was first investigated. A number of all carbon five-membered rings bearing a scope of *para*-, *meta*- as well as *ortho*-substituted aryl rings were smoothly

**Fig. 2** Substrate scope of the cyclopentanone oxime esters **1**. Reaction conditions: **1** (0.10 mmol), TMSCN (1.5 equiv), Cu(MeCN)$_4$PF$_6$ (2 mol %), **L1** (3 mol %), Ir(ppy)$_3$ (0.5 mol%) in DMAc (1.0 mL) at RT for 36 h under the irradiation of 5 W blue LEDs. [a]1.0 mmol scale. [b]Cu(MeCN)$_4$PF$_6$ (1.5 mol%), **L1** (2.25 mol%). TMS Trimethylsilyl. Ts $p$-Methyl benzenesulfonyl.

opened for asymmetric cyanation in high enantioselectivity (84–92% *ee*). Meanwhile, both electron-donating groups such as phenyl (**1a**), methyl (**1c**), *tert*-butyl (**1d**), trimethylsilyl (**1e**), methoxy (**1f**), phenoxy (**1g**), methylthio (**1h**), amino (**1i**), and electron-withdrawing groups such as fluoro (**1j**), chloro (**1k**), trifluoromethoxy (**1l**), trifluoromethyl (**1m**, **1p**), ester (**1n**, **1q**), cyano (**1o**) were all tolerated under this mild catalytic system. Of note is that the tolerance of halogens (**2j**, **2k**) enabled further synthetic elaboration of such chiral dinitriles via known transition-metal-catalyzed cross-coupling reactions. To our satisfaction, such asymmetric ring-opening cyanation reactions of 2-naphthyl, 1-naphthyl, fluorenyl-derived substrates also proceeded smoothly to give the desired products **2r**, **2s**, **2t** with good

yields and high *ee* values. Remarkably, the cyclopentanone oxime esters bearing heteroarenes, including indolyl (**1u**) and benzothiophenyl (**1v**), were also compatible with this catalytic system, albeit in slightly lower yields.

**Substrate scope of the cyclobutanone oxime esters 3**. After the photoredox and copper-catalyzed ring-opening cyanation of cyclopentanone oxime esters has been established, we next set out to explore the transformations of cyclobutanone derivatives for enantioselective synthesis of 2-arylpentanedinitriles. After careful screening of reaction conditions (Supplementary Tables 4–6), as expected, the asymmetric ring-opening cyanation of oxime ester **3**

**Fig. 3** Substrate scope of the cyclobutanone oxime esters **3**. Reaction conditions: **3** (0.10 mmol), TMSCN (1.5 equiv), CuSCN (3 mol%), **L1** (3.6 mol%) in acetone (1.0 mL) at 10 °C for 24 h. [a]CuSCN (1 mol%), **L1** (1.2 mol%). [b]48 h. [c]1.0 mmol scale. [d]CuSCN (5 mol%), **L1** (6 mol%). [e]ee of the crude product.

could be accessed by copper catalysis alone, presumably because higher inclination of strain release of four-membered ring provided major driving force for C–C single bond cleavage. Since the reaction could be performed under even lower temperature (10 °C), the enantioselectivity was further improved (up to 96% *ee*), and more active substituents could be well tolerated in this transformation. As shown in Fig. 3, not only relatively inactive halides including F and Cl, but also more active halides including Br and I were compatible with this catalytic reaction, which clearly broaden synthetic potential of such chiral dinitriles for further transformations (**3i–3l**). Additionally, both electron-rich and electron-deficient substituents were well compatible with this reaction system (**3a–3h**, **3m–3r**). Notably, cyclobutanone oxime

esters installed with fused rings including 1-naphthyl and 2-naphthyl, as well as heteroarenes including benzothiophenyl, indolyl and thiazolyl were also suitable in this asymmetric reaction to give corresponding products with good results (**3s-3w**). To our delight, the reactions proceeded smoothly with good efficiency and excellent enantioselectivity even if copper catalyst loading was decreased to 1 mol% (**4c**, **4h**, **4k**, **4p**, **4r**, and **4s**). The absolute configuration of **4g** was determined to be *R* by X-ray crystallography.

**Synthetic applications**. To demonstrate the utility of this methodology, further derivatizations of the chiral dinitriles were next

**Fig. 4** Synthetic applications. **a** 2a or 4 h was converted to chiral diamides. **b** 2a or 4 h was converted to chiral diamines. Boc₂O Di-*tert*-butyl decarbonate.

**Fig. 5** Preliminary mechanistic studies. **a** Radical trapping experiment with TEMPO. **b** Radical trapping experiment with O₂. TEMPO 2,2,6,6-Tetramethylpiperidinooxy.

carried out (Fig. 4). Firstly, treatment of the chiral dinitriles **2a** and **4h** with concentrated sulphuric acid in AcO*t*Bu, known as a modified Ritter reaction, afforded the corresponding *N-tert*-butyl diamides **5** and **6**, respectively, in good yields (86%, 84%) with excellent enantioselectivity (90% *ee*, 94% *ee*)[67]. Moreover, reduction of chiral **2a** and **4h** by BH₃-THF complex followed by treatment with (Boc)₂O furnished Boc-protected diamines **7** and **8**, respectively, in good yields (79%, 70%) without obvious loss of enantioselectivity[65].

**Mechanistic studies**. To gain some insights into mechanism of the reactions, several control experiments were conducted (Fig. 5). Firstly, the subjection of 1.0 equiv of TEMPO to the standard conditions for **1a** and **3h** could completely inhibited the desired reactions, and the corresponding TEMPO-trapped products **9** and **10** were isolated in 58 and 88% yields, respectively. Meanwhile, when the reactions of **1a** and **3h** were carried out under air atmosphere, both gave only cyano-containing ketones **11** and **12**, respectively, (29 and 57% yields) and no desired dinitriles was obtained. All these results clearly indicated that benzylic radicals were involved in both catalytic cycles.

Based on the above observations and previous reports[60–66,68,69], the proposed mechanisms of two catalytic processes were depicted in Fig. 6. As for cyclobutanone oxime esters **3**, initially,

substrates **3** would undergo SET (single electron transfer) with L*Cu(I)(CN) in the presence of TMSCN affording L*Cu(II)(CN)₂ and iminyl radical **I**, which generated benzylic radical **II** via a radical-mediated C–C bond cleavage. The capture of radical **II** by L*Cu(II)(CN)₂ to give Cu(III) species **III**, followed by reductive elimination furnished dinitriles **4** and regenerated the catalyst L*Cu(I)(CN) (Cycle a, Fig. 6). Meanwhile, as for cyclopentanone oxime esters **1**, the catalytic cycle regarding a dual photoredox/copper catalysis was described as cycle b. Firstly, the SET between oxime esters **1** and the excited state of photocatalyst Ir(III)* provided iminyl radical **IV** under irradiation of blue LEDs, and then the resulting Ir(IV) species oxidized L*Cu(I)(CN) to L*Cu(II)(CN)₂ with participation of TMSCN. The benzylic radical **V**, which was generated by the C–C bond cleavage of iminyl radical **IV**, could be trapped by L*Cu(II)(CN)₂ to deliver Cu(III) species **VI**. Final reductive elimination of intermediate **VI** would give desired products **2** and regenerate the Cu(I) catalyst. Definitely, the catalytic process enabled by single copper catalysis could not be excluded in the ring-opening reaction of cyclopentanone oxime esters (entries 18–19, Table 1). Of note is that the Cu(I)/Cu(II) catalytic cycle involving direct cyano group transfer from chiral copper cyanide could not be excluded for both catalytic processes at this stage.

**Fig. 6** Proposed mechanisms. **a** copper catalysis. **b** a dual photoredox/copper catalysis.

## Discussion

In conclusion, we have developed an enantioselective ring-opening cyanation of cycloketone oxime esters for facile access to chiral 1,5- and 1,6-dinitriles. Specifically, C–C single bonds in less strained 5-membered cycloketone oxime esters can be cleaved for asymmetric cyanation via a radical pathway by combining photoredox and copper catalysis. And, as for strained cyclobutanone oxime esters, asymmetric ring-opening cyanation can be realized by single copper catalysis. Both reactions feature mild conditions, high catalytic reactivity, excellent enantioselectivity, low catalyst loading and broad substrate scope. This radical relay strategy offers a solution for remote functionalization by ring-opening C–C single bonds cleavage, and provides an efficient approach for rapid formation of chiral dinitriles. The synthetic utility of the protocols has been demonstrated by further transformations of the chiral dinitriles to diamides and diamines without obvious loss of enantioselectivity. Further exploration on enantioselective remote difunctionalization via C–C single bonds cleavage are currently ongoing in our laboratory.

## Methods

**Procedure for the asymmetric ring-opening cyanation of oxime esters 1**. To a 25 mL Schlenk tube, Cu(CH$_3$CN)$_4$PF$_6$ (3.7 mg, 0.01 mmol), chiral bisoxazoline ligand **L1** (5.3 mg, 0.015 mmol), Ir(ppy)$_3$ (1.6 mg, 0.0025 mmol) were added in super dry DMAc (5 mL) under N$_2$ atmosphere. The tube was sealed with a Teflon-lined cap, then the mixture was stirred at room temperature for 0.5 h. The catalyst solution I was prepared firstly and used in the next step.

To a 25 mL Schlenk tube containing substrate **1** (0.1 mmol, 1.0 equiv), catalyst solution I (1 mL) and TMSCN (0.15 mmol, 1.5 equiv) were sequentially added under N$_2$ atmosphere. The tube was sealed with a Teflon-lined cap, and the mixture was stirred at a distance of ~5 cm from a 5 W blue LEDs at room temperature for 36 h. The reaction mixture was diluted with ethyl acetate (10 mL). The organic layer was washed with brine (3 × 5 mL) and dried over anhydrous Na$_2$SO$_4$. After filtration and concentration, the residue was purified by silica gel chromatography with petroleum ether and ethyl acetate (PE/EA = 9:1~3:1) to afford the product **2**.

**Procedure for the asymmetric ring-opening cyanation of oxime esters 3**. To a 5 mL Schlenk tube, CuSCN (1.8 mg, 0.015 mmol), chiral bisoxazoline ligand **L1** (6.4 mg, 0.018 mmol) were added in degassed acetone (1 mL) under N$_2$ atmosphere. The tube was sealed with a Teflon-lined cap, then the mixture was stirred at room temperature for 0.5 h. The catalyst solution II was prepared firstly and used in the next step.

To a 25 mL Schlenk tube containing substrate **3** (0.1 mmol, 1.0 equiv), 0.8 mL acetone, catalyst solution II (200 uL) and TMSCN (0.15 mmol, 1.5 equiv) were sequentially added under N$_2$ atmosphere. The tube was sealed with a Teflon-lined cap, and the mixture was stirred at 10 °C for 24 h. Then solvent was removed under vacuum and the residue was purified by silica gel chromatography with petroleum ether and ethyl acetate (PE/EA = 10:1~5:1) to afford the product **4**.

## Data availability

The authors declare that all the data supporting the findings of this research are available within the article and its supplementary information. The characterization data (Supplementary Figs. 1–268), the optimization of reaction conditions (Supplementary Tables 1–6) and the crystallography data (Supplementary Tables 7–12, Supplementary Fig. 269) are available within Supplementary Information. The crystallography data have been deposited at the Cambridge Crystallographic Data Center (CCDC) under accession number CCDC: 1911616 (**4g**) and can be obtained free of charge from www.ccdc.cam.ac. uk/data_request/cif.

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

## Acknowledgements

We gratefully acknowledge the Strategic Priority Research Program of the Chinese Academy of Sciences (Grant No. XDB20000000), the National Basic Research Program of China (973 Program 2015CB856600), the National Science Foundation of China (21971228, 21772187, 21522208) for financial support.

## Author contributions

T.W. designed and performed the experiments. Y.-N.W., R.W., B.-C.Z., C.Y., and Y.-L.L. helped to complete the experiments. X.-S.W. directed the project and wrote the manuscript. All authors interpreted the results on the manuscript.

## Competing interests

The authors declare no competing interests.
