## [Peer Review File · Nature Communications]

Reviewers' comments:

Reviewer #1 (Remarks to the Author):

Since the pioneering work of Zard in 1990s (J. Am. Chem. Soc. 1991, 113, 1055), transition-metal or photoredox catalysis-promoted C–C bond β -scission reactions of iminyl radical from cycloketoxime esters have proven to be an efficient access to distal cyano-substituted alkyl radicals. These intermediates has been broadly applied in the construction of C–C and C–X (X= O, S, Se, Te, B, N) bonds in recent 3 years. However, an asymmetric version has not been hitherto disclosed. In the submitted manuscript, Wang and co-workers described the first copper-catalyzed enantioselective cyanation of cycloketoxime esters with TMSCN to produce chair 1,3- and 1,4-dinitriles. The resultant dinitriles can be further converted into the corresponding optically active diamides and diamines. This work, in my opinion, is a breakthrough in cyanoalkyl radical chemistry, and represents a trend for the transformations involving cycloketoxime esters. Thus, this reviewer think that this manuscript can be published on Nature Communications after the following issues are solved.

- 1) In the abstract and introduction part, the authors overemphasized the difference of intrinsic ring strain between 3- or 4-membered rings and currently used 5-membered cycloketoxime esters. I would think it's not suitable. Firstly, most cases of ring-opening of 3- or 4-membered rings undergo ionic pathway, which, in a way, can't compared to a radical way. Then, the intrinsic strain release should also be benefit for the C–C bond β -scission process of cyclopentanone oxime esters. If not, the 6-, 7-, 8-... membered cycloketoxime esters should all be suitable for this reaction. The utilization of ring-opening of cyclopentanone oxime esters has been widely reported, it is not the key advance for this work. So the authors should modify the abstract and introduction part.
- 2) In previous work of cycloketoxime esters, a Cu(I)/Cu(II) catalytic cycle is generally proposed, while Cu(I)/Cu(III) cycle appears only in a few cases (Angew. Chem. Int. Ed. 2018, 57, 15505; Chem. Commun. 2019, 55, 5347). This work gives a direct evidence for the formation of a high-valent alkyl Cu(III) complex. This should be stated in the main text.
- 3) The substrate scope seems to be limited. What about the α -alkyl or para-substituted cyclopentanone or cyclobutanone oxime esters? Give several successful cases or failures.
- 4) In SI, the ¹³C of compound 9 is in low quality. Please do a new one to replace it.
- 5) A 1.0 mmol scale experiment should be carried out in either main text or SI.

Reviewer #2 (Remarks to the Author):

This manuscript described the facile access to chiral 1,3- and 1,4-dinitriles through radical-mediated C–C bond cleavage and enantioselective cyanation from four- and five-membered cycloketone oxime esters. This enantioselective cyanation reaction represents the first example of asymmetric radical transformation of cycloketone oxime ester. Moreover, asymmetric reactions involving radical-promoted ring-opening C–C bond cleavage are rather limited, therefore, the asymmetric transformations shown in this manuscript are really intriguing. Due to the difference of ring strain, two catalytic systems, single copper catalysis and dual photoredox/copper catalysis, were applied to the ring-opening reactions of four- and five-membered cycloketone oxime esters respectively and both reactions proceeded under mild conditions to afford chiral dinitriles with high yields and excellent enantioselectivity. The reactions also featured low catalyst loading and broad substrate scope. It is worth mentioning that the catalyst loading could be as low as 1 mol% for the reactions of several cyclobutanone oxime esters. The reported transformations provided an effective method to synthesize chiral dinitriles, which are widely used in many fields including polymer materials, enzyme catalysis and organic synthesis. The practical utility of this method has been demonstrated by conversion of the chiral dinitriles to valuable optically active diamides and diamines. Mechanistically, the involvement of benzylic radical intermediate was verified by the radical trapping experiments. Take all this together, I clearly support its publication in Nature Communications. Meanwhile, some minor issues should also be considered, as shown below:

- (1) The ring-opening reactions of four- and five-membered cycloketone oxime esters had been achieved. What about six or more-membered cycloketone oxime esters?
- (2) As shown in supplementary table 1, in the ring-opening reaction of cyclopentanone oxime esters, the yield increased obviously while the copper catalyst loading decreased gradually from 10 mol% to 2 mol%. Can you explain that?
- (3) As shown in Fig. 2, several electron-deficient substituents installed on meta-position of phenyl ring were tolerated in the ring-opening reaction of cyclopentanone oxime esters. What about para-substituted electron-deficient group such as ester?
- (4) As to the product 4o, ee value present in the table was determined from the crude product. What is result if ee value is determined after being isolated?
- (5) The ring-opening reaction of cyclopentanone oxime esters could be realized by single copper catalysis. By comparison, what are the advantages about employment of dual photoredox/copper catalysis?
- (6) Page 5: In the sentence "photoredox catalytic system and copper catalysis are both essential for this highly efficient transformation." "are" should be corrected to "were".
- (7) Page 8: In the sentence "benzothiophenyl, indolyl and thiazolyl were also suitable substrates in this asymmetric reaction to give corresponding products", "substrates" should be deleted.
- (8) Page 9: In the sentence "in the presence of TMSCN afforded L^{*}Cu(II)(CN)₂ and iminyl radical I", "afforded" should be corrected to "affording".
- (9) Page 9: In the sentence "and then the resulting Ir(IV) species oxidizes L^{*}Cu(I)(CN)", "oxidizes" should be corrected to "oxidized".
- (10) In supplementary information, as for ¹³C NMR of substrate 3d, "125.7 (d, J = 3.7 Hz)" should be corrected to "125.7 (q, J = 3.7 Hz)"; as for ¹H NMR of product 4k, "7.68 – 7.46 (m, 3H)" should be corrected to "7.68 – 7.46 (m, 2H)".

Reviewer #3 (Remarks to the Author):

This manuscript by Wang and coworkers described a radical-mediated ring opening of cycloketone oxime esters followed by enantioselective cyanation to construct chiral dinitriles, which are highly valuable products for organic synthesis. In contrast to the previous reports, this method extended the asymmetric ring-opening and radical transformations of cycloketone oxime ester to the four- and less-strained five-membered rings. Both reactions feature mild conditions, high catalytic reactivity, excellent enantioselectivity, low catalyst loading and broad substrate scope. In general, I recommend this work to be published in Nature Communications after the following minor revisions:

- (1) Besides cyclobutan- and cyclopentanone oxime esters, did the authors try other less-strained rings?
- (2) In table 1, the footnote "d" is unnecessary, as there is already a column indicating the reaction temperature.
- (3) I strongly suggested the authors to be check the structures of all their products, as all the "R = p-CF₃C₆H₄" was miswritten into "R = p-CF₃C₆H₅".
- (4) I suggested the authors to number the footnote according to the order that they appear in figure 3. In addition, for product 4o, why did the authors determine the ee value of the crude product but not the purified product?
- (5) For the sake of reproductively of this protocol, we suggested the authors adding the pictures of the photoreactions' set-up and the reaction performed in dark in the supporting information so that other scholars can achieve the similar results as the manuscript demonstrated.
- (6) In supplementary information, some NMR data are incorrect and thus need to be revised. Concerning ¹³C NMR of substrate 3d, 3o, 2d, the revisions are as follow, 3d: "125.7 (d, J = 3.7 Hz)" should be revised to "125.7 (q, J = 3.7 Hz). 3o: there are two same peaks"130.1, 130.1", this should also be revised. 2d:"77.5, 77.2, 76.8" should be deleted.
- (7) Some grammatical mistakes are shown below:
 - (a) Page 5, line 21: "are" should be revised to "were".
 - (b) Page 8, line 6: "thus" should be deleted.

- (c) Page 9, line 12: "afforded" should be revised to "affording".
- (d) Page 9, line 19: "oxidizes", should be revised to "oxidized".

Reviewer 1:

1. In the abstract and introduction part, the authors overemphasized the difference of intrinsic ring strain between 3- or 4-membered rings and currently used 5-membered cycloketoxime esters. I would think it's not suitable. Firstly, most cases of ring-opening of 3- or 4-membered rings undergo ionic pathway, which, in a way, can't be compared to a radical way. Then, the intrinsic strain release should also be beneficial for the C–C bond β -scission process of cyclopentanone oxime esters. If not, the 6-, 7-, 8-... membered cycloketoxime esters should all be suitable for this reaction. The utilization of ring-opening of cyclopentanone oxime esters has been widely reported, it is not the key advance for this work. So the authors should modify the abstract and introduction part.

Response: Thanks for the discussion. Compared with substrates containing 4-membered rings, the asymmetric ring-opening cyanation of less strained cyclopentanone oxime esters is more difficult. Indeed, the ring-opening cyanation of cyclopentanone oxime esters could be realized by single copper catalysis, but giving relatively lower yield while a higher reaction temperature (50 °C) and a higher catalyst loading (10 mol%) were required. In contrast, dual photoredox/copper catalysis could reduce the reaction temperature to ambient RT, which was beneficial for enantioselectivity control and functional group tolerance. Moreover, the yield could be higher and the copper catalyst loading could be as low as 2 mol%.

However, to avoid any misunderstanding, we have modified the introduction and abstract accordingly, as suggested and shown below:

- a) Introduction, “However, most of these reactions via ionic pathway focused on C–C bond cleavage of small ring systems”.
 - b) Fig. 1, “**d** This work: asymmetric radical transformations of four- and five-membered rings.”.
 - c) Abstract, “Employment of dual photoredox/copper catalysis is essential for the asymmetric ring-opening cyanation of cyclopentanone oxime esters.”.
2. In previous work of cycloketoxime esters, a Cu(I)/Cu(II) catalytic cycle is generally proposed, while Cu(I)/Cu(III) cycle appears only in a few cases (Angew. Chem. Int. Ed. 2018, 57, 15505; Chem. Commun. 2019, 55, 5347). This work

gives a direct evidence for the formation of a high-valent alkyl Cu(III) complex. This should be stated in the main text.

Response: Thanks for the suggestion. While we incline to propose Cu(I)/Cu(III) cycle for both transformations, however, the Cu(I)/Cu(II) catalytic cycle involving direct cyano group transfer from chiral copper cyanide in an enantioselective way could not be excluded at this stage. We have added the following sentence into the part of Mechanistic studies, Page 6 in the manuscript, as “Of note is that the Cu(I)/Cu(II) catalytic cycle involving direct cyano group transfer from chiral copper cyanide could not be excluded for both catalytic processes at this stage.”

The two mentioned references have been cited as references 68 and 69 in the manuscript, as suggested.

3. The substrate scope seems to be limited. What about the α -alkyl or para-substituted cyclopentanone or cyclobutanone oxime esters? Give several successful cases or failures.

Response: Thanks for the nice advice. The subsection of α -Bn cyclopentanone oxime esters into the standard conditions gave no product, while the reaction of α -Bn cyclobutanone oxime ester afforded the desired product with 82% yield and 23% ee. Both results have been added into the supporting information, as shown in Supplementary Figure 270.

Several cyclopentanone oxime esters with electron-deficient substituents installed on the para-position have been tested under the standard conditions, which

showed ester and CF₃ groups could afford relatively lower ee and moderate yield while NO₂ gave only trace amount of the product. All these results have been added into the manuscript and the supporting information, shown as **2p** and **2q** in Fig. 2 in the manuscript, and **2w** in the Supplementary Figure 270 in the supporting information.

4. In SI, the ¹³C of compound **9** is in low quality. Please do a new one to replace it.

Response: A new ¹³C-NMR figure of compound **9** has been collected and displayed in SI, as suggested.

5. A 1.0 mmol scale experiment should be carried out in either main text or SI.

Response: Thanks for the suggestion. The 1.0 mmol scale experiments using **1a** and **3h** as the substrate have been performed, as suggested. The results were added into Fig. 2 and Fig. 3 in the manuscript and marked in red.

Reviewer 2:

1. The ring-opening reactions of four- and five-membered cycloketone oxime esters had been achieved. What about six or more-membered cycloketone oxime esters?

Response: Thanks for the suggestion. Six and seven-membered cycloketone oxime esters have also been investigated under the standard conditions, but both gave none of the desired product.

2. As shown in supplementary table 1, in the ring-opening reaction of

cyclopentanone oxime esters, the yield increased obviously while the copper catalyst loading decreased gradually from 10 mol% to 2 mol%. Can you explain that?

Response: As depicted in Fig. 6b, both Ir(III)/ Ir(IV) and Cu(I)/Cu(III) cycle were involved in the catalytic process of ring-opening asymmetric cyanation of cyclopentanone oxime esters. It could be explained by the rate match of two cycles in the transformation, that is the gradual decrease of copper catalyst loading from 10 mol% to 2 mol% would reduce the rate of Cu(I)/Cu(III)-catalyzed cyanation to match the rate of photocatalytic ring-opening process, and thus improve the reaction efficiency.

3. As shown in Fig. 2, several electron-deficient substituents installed on meta-position of phenyl ring were tolerated in the ring-opening reaction of cyclopentanone oxime esters. What about para-substituted electron-deficient group such as ester?

Response: Thanks for the advice. Several cyclopentanone oxime esters with electron-deficient substituents installed on para-position have been tested under the standard conditions, which showed ester and CF₃ groups could afford lower ee and moderate yield while NO₂ gave only trace amount of the product. All these results have been added into the manuscript, shown as **2p** and **2q** in Fig. 2 in the manuscript, and **2w** in the Supplementary Figure 270 in the supporting information.

4. As to the product 4o, ee value present in the table was determined from the crude product. What is result if ee value is determined after being isolated?

Response: Actually, product **4o** racemized partly during the purification by silica gel chromatography, which gave a sharp decline of ee to only 28%. This result has been added into Page S167 in the supporting information.

5. The ring-opening reaction of cyclopentanone oxime esters could be realized by single copper catalysis. By comparison, what are the advantages about employment of dual photoredox/copper catalysis?

Response: Indeed, the ring-opening cyanation of cyclopentanone oxime esters could be realized by single copper catalysis, but giving relatively lower yield while a higher reaction temperature (50 °C) and a higher catalyst loading (10 mol%) were required. In contrast, dual photoredox/copper catalysis could reduce the reaction temperature to ambient RT, which was beneficial for enantioselectivity control and functional group tolerance. Moreover, the yield could be higher and the copper catalyst loading could be as low as 2 mol%.

6. Page 5: In the sentence “photoredox catalytic system and copper catalysis are both essential for this highly efficient transformation.” “are” should be corrected to “were”.

Response: The mistake has been corrected, as suggested.

7. Page 8: In the sentence “benzothiophenyl, indolyl and thiazolyl were also suitable substrates in this asymmetric reaction to give corresponding products”, “substrates” should be deleted.

Response: The mistake has been corrected, as suggested.

8. Page 9: In the sentence “in the presence of TMSCN afforded L*Cu(II)(CN)₂ and iminyl radical I”, “afforded” should be corrected to “affording”.

Response: The mistake has been corrected, as suggested.

9. Page 9: In the sentence “and then the resulting Ir(IV) species oxidizes L*Cu(I)(CN)”, “oxidizes” should be corrected to “oxidized”.

Response: The mistake has been corrected, as suggested.

10. In supplementary information, as for ¹³C NMR of substrate 3d, “125.7 (d, J = 3.7 Hz)” should be corrected to “125.7 (q, J = 3.7 Hz)”; as for ¹H NMR of product 4k, “7.68 – 7.46 (m, 3H)” should be corrected to “7.68 – 7.46 (m, 2H)”.

Response: The mistakes have been corrected, as suggested.

Reviewer 3:

1. Besides cyclobutan- and cyclopentanone oxime esters, did the authors try other

less-strained rings?

Response: Some other less-strained rings, including 6-, and 7-membered cycloketone oxime esters have been tested under the standard conditions, but affording none of the desired product.

- In table 1, the footnote “d” is unnecessary, as there is already a column indicating the reaction temperature.

Response: Thanks for the advice. The mistake has been corrected, as suggested.

- I strongly suggested the authors to be check the structures of all their products, as all the “R = *p*-CF₃C₆H₄” was miswritten into “R = *p*-CF₃C₆H₅”.

Response: All these structures have been double-checked, and the mistake has been corrected accordingly.

- I suggested the authors to number the footnote according to the order that they appear in figure 3. In addition, for product 4o, why did the authors determine the ee value of the crude product but not the purified product?

Response: Thanks for the suggestion. The footnotes in figure 3 have been numbered according to the order that they appear, as suggested.

Meanwhile, product **4o** racemized partly during the purification by silica gel chromatography, which gave a sharp decline of ee to only 28%. This result has been added into Page S167 in the supporting information.

- For the sake of reproductively of this protocol, we suggested the authors adding the pictures of the photoreactions’ set-up and the reaction performed in dark in the supporting information so that other scholars can achieve the similar results as the manuscript demonstrated.

Response: Thanks for the nice suggestion, and the pictures of such reactions have been added into the supporting information, as shown in Page S154.

- In supplementary information, some NMR data are incorrect and thus need to be revised. Concerning ¹³C NMR of substrate 3d, 3o, 2d, the revisions are as follow,

3d: “125.7 (d, J = 3.7 Hz)” should be revised to “125.7 (q, J = 3.7 Hz). 3o: there are two same peaks“130.1, 130.1”, this should also be revised. 2d:“77.5, 77.2, 76.8” should be deleted.

Response: The mistakes have been corrected, as suggested.

7. Some grammatical mistakes are shown below:

- (a) Page 5, line 21: “are” should be revised to “were”. (b) Page 8, line 6: “thus” should be deleted. (c) Page 9, line 12: “afforded” should be revised to “affording”.
- (d) Page 9, line 19: “oxidizes”, should be revised to “oxidized”.

Response: The mistakes have been corrected, as suggested.

REVIEWERS' COMMENTS:

Reviewer #1 (Remarks to the Author):

The authors have made a good revision. I think the manuscript can be accepted by Nat. Commun. with current version.

Reviewer #2 (Remarks to the Author):

The authors has revised this manuscript according the reviewers' comments. The revised manuscript can be accepted for publication as its current form.

Reviewer #3 (Remarks to the Author):

I feel that the points raised in the previous round of review have been satisfactorily addressed.